# History and Toxinology of Palytoxins

**DOI:** 10.3390/toxins16100417

**Published:** 2024-09-26

**Authors:** Harriet L. Hammond, Chad J. Roy

**Affiliations:** 1Center for Airborne Infection & Transmission Science, Tulane University School of Medicine, New Orleans, LA 70112, USA; hhammond1@tulane.edu; 2Division of Microbiology, Tulane National Primate Research Center, Covington, LA 70433, USA

**Keywords:** palytoxins, ovatoxins, zoanthids, aerosol, inhalation, oral exposure, disease models

## Abstract

Palytoxins are a group of highly potent and structurally complex marine toxins that rank among some of the most toxic substances known to science. Palytoxins are naturally synthesized by a variety of marine organisms, including *Palythoa* zoanthids, *Ostreopsis* dinoflagellates, and *Trichodesmium* cyanobacteria, and are widely distributed in tropical and temperate regions where they can bioaccumulate in marine life. The evolution of research on palytoxins has been an intricate exchange between interdisciplinary fields, drawing insights from chemistry, biology, medicine, and environmental science in efforts to better understand and mitigate the health risks associated with this family of toxins. In this review, we begin with a brief history covering the discovery of this group of toxins and the events that led to its isolation. We then focus on the chemical structure of these compounds and their proposed mechanism of action. Finally, we review in vitro, ex vivo, and in vivo studies related to their toxicity, with the aim to provide a broad overview of the current knowledge on palytoxin toxinology.

## 1. Introduction

The primary producers of palytoxin and its various analogues and congeners include zoanthid coelenterates (for example, corals) of the genera *Palythoa* and *Zoanthus*, certain species of dinoflagellates of the genus *Ostreopsis*, and filamentous cyanobacteria of the genus *Trichodesmium* [1,2,3,4,5,6,7,8]. Palytoxins play a multifaceted ecological role in these marine organisms, serving as a chemical defense mechanism to help deter predators and inhibit the growth of competing microorganisms in densely populated marine environments, and as an adaptive mechanism in response to environmental stressors such as changes in temperature, salinity, acidification, nutrient availability, or bleaching events, helping the primary producers survive and thrive in varying conditions within marine ecosystems. Palytoxins exhibit a broad distribution and can be found in both tropical and temperate regions. These toxins can bioaccumulate in various marine organisms, including fish, crabs, mussels, sponges, and cnidaria that live near or feed on palytoxin-producing corals and algae, facilitating the trophic transfer of palytoxin through the food web [9,10,11,12,13,14]. Harmful algal blooms involving *Ostreopsis* and *Trichodesmium* species can release palytoxin into the surrounding water, potentially disrupting local marine ecosystems and leading to significant ecological impacts [14,15,16,17,18,19]. Accounts of poisonings and exposures to palytoxins are being reported with increasing frequency throughout the world, particularly of people who have ingested contaminated seafood; of dermal, ocular, and inhalational exposures of aquarium hobbyists who have handled *Palythoa* corals; and of inhalational and cutaneous exposures of beachgoers to marine aerosols during *Ostreopsis* microalgal blooms [17,20,21,22,23].

## 2. History

According to Hawaiian folklore, the native people of the Hawaiian Islands used exudates of a rare, yet very toxic “limu-make-o-Hana” (the deadly seaweed of Hana) to impart lethal potency to the tips of spears and arrows used for warfare. In 1971, researchers at the University of Hawaii followed up on this precedent, which eventually led to the isolation of palytoxin from the cnidarian zoanthid *Palythoa toxica* located in Hana Bay on the island of Maui [21]. The work initiated by Moore and Scheuer is of historical significance and has laid the foundation for understanding the chemistry and biology of marine toxins, leading to the discovery of the biodiversity of new zoanthid species and subsequent analyses of palytoxins [24]. Referenced in their work is a dissertation by David Henry Attaway submitted in 1968 as part of the requirements for the Doctor of Philosophy degree at the University of Oklahoma. Attaway’s dissertation focuses on the purification and characterization of palytoxin extracted from *Palythoa caribaeorum* collected at a reef near Port Royal, Jamaica [25]. During this same time period, in 1969, while Japanese researchers at the University of Tokyo were investigating the cause of ciguatera poisoning—a type of food poisoning caused by eating fish contaminated with ciguatoxin and maitotoxin from the dinoflagellate *Gambierdiscus toxicus*—they incidentally discovered *Palythoa tuberculosa* in the digestive tract of the poisonous filefish *Alutera scripta* that had been collected from an island in the southern part of Japan [26]. This new marine toxin was designated as “aluterin” but was soon found to have the same chemical properties as palytoxin isolated by Scheuer from other *Palythoa* spp. [26,27]. In 1981, the Hawaiian and Japanese research groups independently went on to describe the molecular structure of palytoxin, both of whom described it as being one of the largest and most complex organic compounds, a fact that still stands to date [28,29].

## 3. Structural Analysis

Palytoxins belong to a group of marine organic molecules known as super-carbon-chain compounds and are recognized as some of the most complex non-proteinaceous biomolecules due to their unique, intricate, linear polyether structures. Contributing to the complexity of palytoxins is their remarkably long 115-carbon straight chain backbone that has many functional groups attached, including cyclic ethers, methyl groups, and hydroxyl groups. Palytoxins are characterized by their high molecular mass and potent biological effects. The molecular weight of palytoxin is approximately 2680 g/mol, far surpassing that of any other natural substance at its time of discovery [30]. It has the molecular formula of C_129_H_223_N_3_O_54_, contains both lipophilic and hydrophilic moieties, and has 64 chiral centers as well as eight double bonds which allow for cis–trans isomerism, contributing to the possibility of more than 10^21^ stereoisomers [1,9,30]. Due to its complexity, the complete stereochemistry of the palytoxin molecule, including absolute configurations, was a significant achievement in the field of chemistry when elucidated in 1982 [28,29,31,32,33]. More than a decade later, in 1994, a team of organic chemists at Harvard University published their work on the total synthesis of palytoxin, marking a significant milestone in understanding the complex chemistry of this unique compound [30,34,35,36]. In 2007 and 2008, researchers provided further insights into the stereochemical configuration of the palytoxin molecule. Employing techniques such as X-ray crystallography and nuclear magnetic resonance, their studies revealed that the palytoxin molecule exists in a dimeric form in an aqueous solution, resembling a pair of horseshoes positioned opposite each other [37,38].

There were slight variations in the structures and molecular weights of the palytoxin molecule reported by the two independent research groups in 1981, suggesting that the preparations obtained from the Hawaiian and Japanese *P. tuberculosa*, Hawaiian *P. toxica*, and Tahitian *Palythoa* sp. samples may have been comprised of several different analogues of palytoxin, each at different ratios [28,29,39,40]. It has since been discovered that the structural composition of palytoxins can vary based on their biological source and geographical origin [9,41,42,43,44,45]. Moreover, the quantity and composition of palytoxins exhibit considerable variations, both seasonal and regional, as well as between and within species, with toxicity fluctuations based on factors such as the bloom phase of dinoflagellates, the particular macroalgal substrate, or the presence of eggs and distribution of female polyps in a coral colony [14,26,45,46,47,48].

Palytoxin isolated from *P. tuberculosa* is considered the analytical standard 2680 Da palytoxin [3,42,49,50,51]. Several structurally related analogues and congeners that share the complex polyether backbone architecture of palytoxin have been identified, differing in aspects such as the quantity and arrangement of hydroxyl and/or methyl groups, as well as chirality; minor variations that could significantly influence their toxicological profile and biological activity [40,41,42,43,44,52,53]. Nearly 20 palytoxin-like compounds are currently known and are organized into four structural families: palytoxins, ostreocins, ovatoxins, and mascarenotoxins [16,52,54]. Despite the observed interspecies and intrastrain variations in the structure of palytoxin analogues and congeners in marine organisms, only a few have been studied and further characterized. For instance, two major palytoxin analogues, isolated from *P. toxica* and *P. tuberculosa* samples collected from the legendary cursed Hawaiian tidepool, were identified as stereoisomers of 42-hydroxy-palytoxin (42*S*-hydroxy-50*S*-palytoxin and 42*S*-hydroxy-50*R*-palytoxin, respectively) [41,42]. Palytoxin and 42-hydroxy-palytoxin have also been found in filamentous cyanobacteria of the genus *Trichodesmium* [7]. Additionally, 42-hydroxy-palytoxin has been isolated and identified in dinoflagellates belonging to the *Prorocentrum* genus, specifically the *P. borbonicum* species [55]. Dinoflagellates belonging to the genus *Ostreopsis* are producers of the palytoxin congeners known as ostreocins, ovatoxins, and mascarenotoxins [9]. The major constituent isolated from cultures of *Ostreopsis siamensis* was determined to be ostreocin-d, definitively identified as 42-hydroxy-3,26-didemethyl-19,44-dideoxypalytoxin with the molecular formula of C_127_H_219_N_3_O_53_ and molecular weight of 2633 Da [2,6,56]. The most abundant toxin produced by the dinoflagellate *O. ovata* is ovatoxin-a; it has a molecular formula of C_129_H_223_N_3_O_52_ and molecular weight of 2648 Da [5,44]. When compared to its parent compound palytoxin, ovatoxin-a has an additional hydroxy group at the 42nd position and is missing three hydroxy groups at the 17th, 44th, and 64th positions, yielding the structural elucidation of 42-hydroxy-17,44,64-trideoxypalytoxin [45,57]. Mascarenotoxins-a and -b are palytoxin congeners present in *O. mascarenensis* which possess a molecular mass in the range of 2500 to 2535 Da [4,58].

The chemical complexity of different palytoxin compounds presents significant challenges to their extraction, separation, identification, and quantitation, even with the use of advanced analytical techniques such as ultra-high-performance liquid chromatography (UHPLC) and high-resolution mass spectrometry (HRMS). Despite these technological advancements, it is important to note that trace amounts of different palytoxin analogues or congeners are invariably present in these preparations, and there is a risk for quantitation errors if the appropriate extraction and analytical methods are not employed. Additionally, palytoxin compounds can be isolated from a variety of biological sources and matrices, each of which can introduce its own unique variables that can potentially influence both the efficiency of the extraction process and reliability of the subsequent analyses. Furthermore, the conditions under which the instruments are used can also have a significant impact on the detection and quantitation of palytoxin compounds. Although specific extraction methods and analytical methodology for palytoxins have been tested and proposed, this is an area where further research and collaboration is urgently needed as variability in the content, purity, and activity of palytoxin preparations poses a significant challenge for reproducibility in subsequent studies [59]. These minor constituents, though often overlooked, can significantly influence the overall biological activity of the palytoxin sample, potentially impacting the outcomes of subsequent toxicity studies.

The only commercially available palytoxin preparation, hereby referred to as palytoxin standard, is offered by FUJIFILM Wako Pure Chemicals Corporation (Osaka, Japan) and its subsidiary Wako Chemicals Europe GmbH (Neuss, Germany) [51]. This preparation, extracted from *P. tuberculosa*, has been fundamental in palytoxin toxicity studies. Reports of its use in scientific investigations appear to begin around 1999, although this date may not be exhaustive [60]. The process of making qualitative and quantitative assessments of palytoxin analogues and congeners assumes that these compounds will exhibit similar behavior to palytoxin standard during chemical analyses due to their structural similarities. It is important to note, however, that the purity of the palytoxin standard preparation itself is not absolute and may vary between batches, with reported purity values ranging between 83 and 95% [49,51,61,62,63,64,65]. These findings highlight the complexity of studying palytoxins and underscore the need for continued advancements in analytical techniques, standardization protocols, and certified reference materials. Novel palytoxin analogues and congeners are still being discovered [66,67,68]. As our understanding of these potent marine toxins continues to evolve, so too will our ability to study them accurately and reliably.

### 3.1. Biosynthesis

While the chemical synthesis of palytoxins has been successfully achieved, the natural biosynthesis of these compounds remains poorly understood. The etiology of palytoxins continues to be a topic of debate within the scientific community, as their biosynthetic pathway and the purported involvement of symbiotic organisms are not yet fully elucidated. While dinoflagellates of the *Ostreopsis* spp. are suspected to be primary producers of palytoxins, and different species of the *Palythoa* and *Zoanthus* genera are known to harbor endosymbiotic dinoflagellates from the family Symbiodiniaceae, which do not produce palytoxins, the role of other potential microbial associates in the biosynthesis of palytoxin has yet to be conclusively determined [48].

Several studies suggest that symbiotic bacteria associated with zoanthids and dinoflagellates may be a source of palytoxins. This is supported by the detection of palytoxin-like hemolytic activity in bacterial extracts from the genus *Pseudomonas*, which were isolated from the dinoflagellate *O. lenticularis* and the Caribbean zoanthids *P. caribaeorum* and *P. mammillosa* [69]. Additionally, similar activity has been found in strains of the *Bacillus cereus* group, as well as the genera *Brevibacterium* and *Acinetobacter*, all isolated from *P. caribaeorum* [70]. Furthermore, after screening 340 isolates from samples of *Palythoa* spp. using a competitive ELISA, it was discovered that Gram-negative bacteria from the *Aeromonas* and *Vibrio* species produce compounds that are antigenically similar to palytoxin [71]. The processes by which these organisms produce toxic polyketides remain poorly understood, highlighting the need for further research to clarify the exact mechanisms and sources involved in the biosynthesis and storage of palytoxins in Zoantharia and Dinoflagellida [72].

### 3.2. Bioaccumulation

Palytoxins, although primarily found in *Palythoa* spp., have been detected in various other marine organisms associated with these zoanthid colonies. These include other soft coral species as well as sponges, gorgonians, sea anemones (*Radianthus macrodactylus*), mussels, and crustaceans like xanthid crabs, which inhabit the crevices of *Palythoa* colonies and have been shown to exhibit palytoxin concentrations similar to those of the zoanthids [11,13,73,74,75]. High concentrations of palytoxins have also been detected in zoanthid predators, such as polychaete worms (*Hermodice carunculate*), crown-of-thorns starfish (*Acanthaster planci*), and Caribbean butterfly fish (*Chaetodon* spp.), which frequently feed on *Palythoa* colonies and prey on crustaceans and worms living in or near these colonies [13,73,74]. Additionally, larger predatory fish species, including filefish, sardines, mackerel, groupers, parrotfish, and barracuda, have been shown to store and accumulate palytoxins in their organs [26,60,76,77,78]. The mechanisms underlying this resistance to its toxic effects remain largely unknown. However, it may involve metabolic pathways that result in the biotransformation of the molecular structure of palytoxin, reducing its toxicity; molecular changes to the Na^+^/K^+^ ATPase that prevent or minimize palytoxin interaction, thereby mitigating its harmful effects; or resistance to palytoxin-induced pore formation in cellular membranes, which could confer protection from the toxin’s disruptive actions. The tolerance to palytoxins in these marine animals may facilitate its biomagnification, transport, and widespread distribution within marine food webs, which poses significant risks to higher trophic levels, including humans.

The detection of palytoxin in edible fish was first reported in 1969, when gut extracts of the filefish *Altera scripta* were found to contain fragments of a *Palythoa* species [26,27]. Reports of palytoxins as the cause of human fatalities and poisoning events have since been documented in various tropical and sub-tropical regions following the consumption of contaminated marine species, including crabs (*Demanlia reynaudii* and *Lophozozymus pictor*) in the Philippines, sardines (*Herklotsichthys quadrimaculatus*) in Madagascar, mackerel (*Decapterus macrosoma*) in Hawaii, triggerfish (*Melichthys vidua*) in Micronesia, and groupers (*Epinephelus* sp.) and blue humphead parrotfish (*Scarus ovifrons*) in Japan [11,12,60,76,77,78,79,80]. In the case of the blue humphead parrotfish, the authors suggested that the palytoxins originated from the microalgal dinoflagellate species *Ostreopsis*, which was found in significant amounts in the gut contents of *S. ovifrons* during a massive bloom in the same district [80]. Clinical symptoms, along with laboratory findings, have indicated palytoxin as the probable causative agent in these fatal and near-fatal cases.

Palytoxicosis resulting from the ingestion of contaminated seafood is characterized by a variety of clinical signs and symptoms, including neurological and gastrointestinal disturbances, as well as significant cardiovascular and respiratory complications. Key clinical manifestations include a bitter or metallic taste, dysgeusia, abdominal spasms, nausea, vomiting, diarrhea, tremors, myalgia, paresthesia, muscle spasms, bradycardia, hypertension, and renal failure. In severe cases, respiratory distress, cyanosis, or coma occurs, with fatal cases typically resulting from myocardial injury [8,9,22,54]. The most common complication of palytoxin poisoning is rhabdomyolysis, a syndrome resulting from direct or indirect muscle injury that leads to the death of muscle fibers. Rhabdomyolysis is characterized by severe myalgia, muscle weakness, muscle breakdown, and the leakage of myocyte contents into the bloodstream. Symptoms of palytoxin-induced rhabdomyolysis typically appear a few hours after ingestion of contaminated seafood and include myocardial damage, marked by elevated levels of myosin light chain, creatine kinase, aspartate aminotransferase, alanine aminotransferase, lactate dehydrogenase, and C-reactive protein, along with altered electrocardiogram readings [1,76,77,78,81,82,83].

Marine biotoxins that cause foodborne illnesses in humans are typically categorized into two groups: water-soluble molecules, such as those responsible for Paralytic Shellfish Poisoning and Amnesic Shellfish Poisoning; and fat-soluble molecules, such as those causing Diarrheic Shellfish Poisoning and Neurotoxic Shellfish Poisoning [19]. However, the complex structure of the palytoxin molecule features both hydrophilic (water-soluble hydroxyl groups) and lipophilic (a fat-soluble long carbon chain) parts, enabling it to interact with various biological membranes and affect multiple systems, which contributes to its high toxicity. In all cases, symptoms overlap and can vary, and may include severe gastrointestinal issues such as diarrhea, nausea, vomiting, and abdominal spasms; neurological conditions like dizziness, ataxia, and partial paralysis; as well as cardiovascular issues and respiratory distress.

Epidemiological studies worldwide have highlighted the potential health risks associated with consuming seafood contaminated with palytoxins, which have been implicated in various foodborne illnesses, including ciguatera, clupeotoxism, and Haff disease [7,60,76,84,85,86,87]. Ciguatera Fish Poisoning is caused by water-soluble maitotoxins and lipid soluble ciguatoxins, both produced by the benthic dinoflagellate *Gambierdiscus toxicus* and transferred through the marine food web [74,88]. Common ciguatoxic fish species are also implicated in palytoxicosis, suggesting that certain cases initially thought to be ciguatera might actually be due to palytoxin-like toxins [76,87,89]. Since ciguatera fish poisoning shares symptoms with other marine food poisonings, atypical symptoms of ciguatera may be caused by palytoxins rather than ciguatoxins. Palytoxin is also implicated in clupeotoxism, a marine poisoning resulting from the consumption of plankton-eating clupeoid fish such as sardines (*Clupeidae*), anchovies, herrings, and mullets [9,60,81,86]. Clupeotoxicosis is associated with blooms of benthic dinoflagellates of the *Ostreopsis* spp., particularly *O. siamensis*, *O. mascarenensis*, *O. lenticularis*, and *O. ovata*, all of which are known producers of palytoxin congeners [5,90,91]. Additionally, palytoxin contamination is associated with Haff disease, a rare syndrome characterized by the acute onset of rhabdomyolysis and elevated serum creatine kinase levels, typically occurring withing 24 h of consuming certain types of fish or crustaceans [83,84,85].

Further research is essential to fully elucidate the role of palytoxins in various marine foodborne illnesses. The widespread presence of palytoxins in diverse marine organisms supports the notion that these toxins are produced by microorganisms and subsequently transmitted through the marine food web via organisms that exhibit resistance or tolerance to their toxic secondary metabolites. For example, moray eels (*Gymnothorax* spp.), as apex predators in the coral ecosystem, are particularly prone to accumulating high levels of ciguatoxins [92]. These marine animals tend to contain more polar congeners of ciguatoxin compared to those produced by the dinoflagellate *Gambierdiscus toxicus*, suggesting that oxidative enzyme systems within the eels modify the less polar congeners into more polar forms [88]. Additionally, in the dinoflagellate *Karenia brevis*, the causative agent of the notorious ‘red tides’ harmful algal blooms and producer of potent neurotoxins known as brevetoxins, RNA of several polyketide synthase (PKS) genes has been localized within their cells or associated bacteria, indicating that both cell types have the genetic capability to produce polyketide-like substances [72]. It remains to be determined if this is also true for palytoxins. Monitoring and managing the presence of palytoxins in marine environments is crucial to prevent incidents of palytoxicosis and ensure seafood safety.

## 4. Mechanism of Action

As the molecular structure of palytoxin was being elucidated, separate research groups were simultaneously engaged in efforts to uncover and comprehend the toxin’s mechanism of action, a process that developed over the course of several years. Early investigations on the pharmacological actions resulting from in vivo and in vitro exposures to palytoxins reveal that it causes notable increases in arterial blood pressure, violent contractions of skeletal, smooth, and cardiac muscles, massive secretions of secretory cells, and depolarization in various types of excitable tissues [27,30]. Moreover, palytoxins lead to the formation of non-selective channels in cell membranes, increasing the permeability of various cations [27,30,93]. Due to its extreme toxicity and the myriad effects the toxin elicits, palytoxin was presumed to function as a cytolysin. Acting on this presumption, Habermann and associates reported in 1981 that palytoxin triggers a delayed hemolysis in erythrocytes, preceded by a rapid, quantifiable loss of potassium [94]. In subsequent investigations, Habermann’s group incorporated the use of ouabain, a cardiac glycoside known for its specific action on the Na^+^/K^+^ ATPase. This approach enabled them to pinpoint the Na^+^/K^+^ ATPase as the precise molecular target of palytoxin [95,96]. Collectively, these findings led Habermann and colleagues to propose that palytoxin interacts with the Na^+^/K^+^ ATPase, transforming it into an ion channel that allows the passive diffusion of cations [97]. In the years that followed, experimental evidence began to accumulate in favor of Habermann’s theory. It was discovered that palytoxin could induce the formation of single channels within a planar lipid bilayer containing purified Na^+^/K^+^ ATPase [98]. Then, in 1997, using synthetic Na^+^/K^+^ ATPase in a cell-free system, it was conclusively proven that this enzyme is the precise target of palytoxin [99].

The Na^+^/K^+^ ATPase is an indispensable, integral membrane protein found in both excitable and non-excitable cells that establishes and maintains the electrochemical gradient. Normal functioning of the Na^+^/K^+^ ATPase, as described by the Post-Albers cycle, is characterized by an energy-dependent process that transports sodium and potassium ions across the cell membrane against their concentration gradients [100,101]. This balance of sodium and potassium levels across the cellular membrane is essential for a variety of physiological processes and is harnessed by various organs to carry out their specific functions [102]. The Na^+^/K^+^ ATPase consists of three subunits: a catalytic, ten-transmembrane α subunit that is responsible for ion transport; a smaller, single-transmembrane β subunit that is involved in ion occlusion; and, wherever expressed, a tissue-specific auxiliary regulatory γ FXYD subunit that modulates the enzyme’s activity [103]. The subunits integrate into the membrane as a functional diprotomeric (αβ)_2_ or triprotomeric (αβ γ)_2_ complex [93,101,104,105]. The pump alternates between two primary conformations, E1 and E2, while progressing through a sequence of four gating reactions [100,101,103,106]. Palytoxin acts on the Na^+^/K^+^ ATPase membrane pump by converting it to an open non-selective cation channel, a modification that leads to numerous downstream cellular effects and a broad spectrum of secondary pharmacological actions [1,30,97,107]. The interaction between palytoxin and Na^+^/K^+^ ATPase occurs in a 1:1 stoichiometry, where a dimer of palytoxin associates with a dimer of Na^+^/K^+^ ATPase [93,108,109].

A recently published study, which utilized the advanced techniques of microcrystal electron diffraction in cryo-electron microscopy along with molecular docking simulations, demonstrated that the hydrophobic region of palytoxin folds into a hairpin motif to attach to the binding pocket of the extracellular gate of the Na^+^/K^+^ ATPase when it is in the E1 state [110]. The binding of palytoxin interferes with the coordinated opening and closing of the outer and inner gates, likely due to steric hindrance, resulting in a conformational change that converts the pump into an open, non-selective monovalent cation channel, causing an efflux of potassium ions and an influx of sodium and other cations across the plasma membrane [30,108,111]. It has been postulated that palytoxin has a preference for binding to the E2P:E2P state, a stage where the extracellular gates of both enzymes are open [93]. However, palytoxin could bind to other states of the ATPase cycle at multiple sites within the pump and with varying degrees of affinity, depending on the phosphorylation state of the ATPase as well as the concentrations of the toxin, Na^+^, and K^+^ ions [93,104,105,112,113]. As the Na^+^/K^+^ ATPase is essential for the maintenance of various cellular functions, destruction of the ion gradient and resting membrane potential of the cell causes a host of deleterious downstream effects, ultimately leading to mitochondrial damage, osmolysis, and necrotic or apoptotic cell death [1,93].

Due to their action on the Na^+^/K^+^ ATPase and the significant alterations they induce in cellular physiology, palytoxins are regarded as some of the most potent marine toxins. They exert toxicity at extremely low concentrations and have been implicated in various health effects, including respiratory distress, cardiovascular malfunction, and neurological sequalae [9]. Case studies on palytoxicosis document incidents resulting from the consumption of contaminated seafood, handling soft corals during maintenance of home aquariums, and encountering aerosolized sea water during toxic algal blooms. In vivo studies demonstrate the acute toxicity of palytoxins administered via various routes of exposure, including intravenous, intraperitoneal, intramuscular, subcutaneous, intratracheal, and oral routes, revealing low LD_50_ values in a broad range of animal species. These studies also highlight potential chronic health effects of sublethal exposure to palytoxins [114]. In vitro and ex vivo studies help elucidate the mechanisms underlying palytoxin-induced cytotoxicity and identify potential targets for therapeutic interventions. Collectively, these investigations underscore the various modalities through which exposure to palytoxins can impact human health.

## 5. In Vitro Studies

In vitro studies are instrumental in elucidating the implications of human exposure to palytoxins. These studies provide a controlled environment to investigate the cellular and molecular mechanisms of palytoxin toxicity, with the MTT assay playing a crucial role. This colorimetric assessment of mitochondrial function serves as a reliable measure of cellular metabolic activity, and by extension, as a proxy for cellular viability. It has been utilized in various cell lines to determine IC_50_ values for palytoxins, representing the concentration at which there is a 50% inhibition of the mitochondrial activity. These in vitro studies are particularly valuable in examining the effects of palytoxin exposure through different routes. For instance, cutaneous exposures, which frequently arise from interactions with corals in domestic aquaria, represent a common route of palytoxin exposure. Investigations employing keratinocytes, such as the HaCaT cell line, provide insight into the dermatological effects of palytoxins, thereby aiding in the development of preventive and therapeutic strategies. The consumption of seafood contaminated with palytoxins is another common exposure route. In vitro studies utilizing intestinal cells, such as the Caco2 cell line or enteric glial cells, elucidate the impact of palytoxin on the gastrointestinal system and its contribution to foodborne illnesses. Inhalational exposures to palytoxins, often originating from algal blooms, pose a significant concern due to the high toxicity of palytoxins. In vitro explorations employing a nervous system cell line, such as Neuro2a, contribute significantly to our comprehension of the neurotoxic effects of palytoxins. Similarly, the incorporation of lung cells, namely A549 and BEAS-2B, in these studies facilitates our understanding of the repercussions of inhalational exposure on respiratory health. In summary, in vitro studies, enhanced by the insights from the MTT assay, are invaluable for understanding the mechanism of action of palytoxins and contribute significantly to elucidating the corresponding toxicokinetics and informing risk assessment. The insights gleaned from the cytotoxic effects of palytoxins across an array of cell types reveal their broad range of pharmacological effects. These investigations serve as a conduit, bridging the divide between observations at the cellular level and real-world human experiences, thereby enhancing our capacity to prevent and manage health issues related to palytoxin exposure.

Increasingly, case reports are documenting instances of dermal toxicity associated with cutaneous exposure to palytoxins. Interactions with palytoxin-producing corals, particularly within the confines of domestic aquariums, can inadvertently lead to toxin contact, resulting in severe inflammatory responses. Clinical manifestations frequently encompass localized erythema, edema, and acute nociception at the site of exposure [1,22,115,116]. The escalating prevalence of these incidents emphasizes the need for heightened awareness and the implementation of rigorous safety protocols among aquarium enthusiasts in efforts to mitigate the risk of dermal toxicity associated with palytoxin exposure. Considering the increasing frequency of such occurrences, scientific investigations have attempted to not only elucidate the cellular processes that drive these dermatoxic reactions but also to quantify the potency of palytoxin in inducing such effects. Keratinocytes, the predominant cell population of the epidermis, play a crucial role in mediating cutaneous inflammatory responses. The HaCaT cell line, an immortalized variant of human keratinocytes, has been extensively employed in skin biology and differentiation studies and, due to their morphological and functional properties which closely resemble those of normal keratinocytes, have also been employed in the screening for cutaneous toxicity. Thus, HaCaT cells present a valuable model for investigating the dermatoxic effects of palytoxin exposure at the cellular level.

Several studies have exposed HaCaT cells to various palytoxin preparations for varying lengths of time, employing the MTT assay to ascertain IC_50_ values and examining the inflammatory responses elicited by the cells. In separate investigations, IC_50_ values were determined in HaCaT cells following a 24 h incubation period with palytoxin standard. One study reported an IC_50_ of 7.71 ng/mL using the MTT assay, while another found an IC_50_ of 34.8 ng/mL using the CCK-8 assay, a method with enhanced sensitivity for assessing cellular metabolic activity [117,118]. In contrast, a third study reported a higher IC_50_ value of 72.4 ng/mL following a shorter, 4 h exposure to the toxin [41]. The extended 24 h incubation period may have facilitated more extensive toxin–cell interactions, potentially leading to lower IC_50_ values. Additionally, the choice of mitochondrial assay and variations in experimental parameters, such as cell density, could also contribute to the observed discrepancies.

Significant cytotoxic variations were observed among analogues and congeners of palytoxin in HaCaT cells. For instance, ovatoxin-d, isolated from cultures of Mediterranean strains of *O.* cf. *ovata*, demonstrated a potency 3.5 times greater than ovatoxin-a, with IC_50_ values approximating 10 ng/mL and 3 ng/mL, respectively, following a 24 h incubation period [117]. Furthermore, the cytotoxicity of palytoxin standard was approximately 10 times greater than that of 42S-hydroxy-50R-palytoxin and 100 times greater than that of 42S-hydroxy-50S-palytoxin, analogues that were isolated from *P. toxica* and *P. tuberculosa*, respectively [41]. Following a 4 h exposure, the IC_50_ for 42S-hydroxy-50S-palytoxin was approximately 2492 ng/mL, while that for 42S-hydroxy-50R-palytoxin was around 268 ng/mL, suggesting that the latter is about 10 times more potent than its stereoisomer in HaCaT keratinocytes [41]. Despite the observed discrepancies in reported IC_50_ values, which could be attributed to differences in exposure times and/or variations in experimental conditions, the collective findings from these studies demonstrate the extreme potency associated with cutaneous exposure to palytoxins, with IC_50_ values in the nanogram range serving as a testament to the toxicity of these compounds. An investigation into the cytotoxic mechanism of palytoxin in HaCaT cells used proteomics and bioinformatics and revealed that palytoxin interacts with the mitogen-activated protein kinase (MAPK) pathway, leading to the down-regulation of MAPK1, MAP2K1, and MAP2K2, which subsequently affects cell proliferation and induces apoptosis [118]. Interestingly, the impact of palytoxin on HaCaT cells extends beyond apoptosis. The study observed an up-regulation of VDAC3, ACSL4, and NCOA4. Based on these findings, it is suggested that palytoxin could also instigate ferroptosis, a distinct form of programmed cell death characterized by iron accumulation and lipid peroxidation [118].

Several studies have demonstrated that exposure to palytoxin standard elicits a pro-inflammatory response in HaCaT keratinocytes. One such study observed a significant increase in interleukin-8 (IL-8) secretion at palytoxin standard concentrations commencing from 5 ng/mL [117]. A concentration of 10 pM (26.8 ng/mL) of palytoxin standard has been associated with the release of IL-6, IL-8, TNF-α, and the inflammatory mediators histamine and prostaglandin-E_2_ [62]. Moreover, THP-1 cells, a widely used human monocytic cell line, exhibited a significant chemotactic response when exposed to conditioned media from palytoxin-treated HaCaT cells. This response was evident in both differentiated (macrophages, immature dendritic cells, and mature dendritic cells) and undifferentiated (monocytes) THP-1 cells. Notably, exposure to the conditioned media did not influence cell proliferation, adhesion, or differentiation [62]. Taken together, these data suggest that keratinocytes play an active role in the recruitment of inflammatory cells following cutaneous exposure to palytoxin, facilitated by the release of pro-inflammatory mediators, and culminating in irritant contact dermatitis rather than sensitization [41,62,117].

While cutaneous exposure to palytoxins presents significant risks, these toxins also pose a threat through another common route: ingestion of contaminated seafood. To gain a deeper understanding of how palytoxins affect the human body when ingested, researchers have turned to in vitro models. The Caco2 human intestinal cell line has proven to be an effective model for the intestinal epithelial barrier due to its ability to resemble mature enterocytes. These cells exhibit key characteristics such as the ability to differentiate and form a monolayer with tight junctions and microvilli, as well as express various enzymes and establish transport systems, making them ideal for studying the absorption and transport of substances across the intestinal epithelial barrier. Numerous studies have utilized Caco2 cells to evaluate the potential toxicity resulting from palytoxin ingestion, focusing on its effects on cell viability, proliferation, and barrier integrity.

One research study assessed the mitochondrial activity of undifferentiated Caco2 cells using the MTT assay and reported an IC_50_ value of 24 ng/L (equivalent to 0.024 ng/mL) following a 4 h exposure to palytoxin standard [119]. Concurrently, the study employed the Sulforhodamine B (SRB) assay, a colorimetric test quantifying cellular protein content, which indicated a decrease in cell density with an IC_50_ value of 53.6 ng/L (equivalent to 0.053 ng/mL) [119]. Another investigation also employed the MTT assay, undifferentiated Caco2 cells, and palytoxin standard to determine IC_50_ values; however, the derived values were notably higher. The IC_50_ values determined by the MTT assay after a 24 h exposure period were 6.7 ng/mL for palytoxin standard, and approximately 18 ng/mL and 6 ng/mL for ovatoxin-a and ovatoxin-d, respectively [117]. Expanding upon the exploration of palytoxin’s effects on Caco2 cells, another study adopted a different approach. This research, which reported higher IC_50_ values than the aforementioned studies, utilized a unique assay and examined additional cellular markers. Specifically, this investigation employed the CyQuant proliferation assay, a fluorescence-based analysis of the total DNA content which serves as an indicator of cellular proliferation, to observe the effects of palytoxin standard on Caco2 cells. A decrease in cellular proliferation was observed at a palytoxin concentration of 0.1 nM (roughly equivalent to 268 ng/mL), with the IC_50_ value determined to be 1 nM (approximately 2680 ng/mL) following a 24 h exposure period. Notably, the study reported complete depolymerization of F-actin, a crucial protein involved in maintaining cellular structure and facilitating movement. Interestingly, other markers typically associated with apoptosis, such as complete mitochondrial depolarization, DNA fragmentation, and caspase activation, were not observed [120]. This suggests that exposure to palytoxin may induce both necrosis and apoptosis in Caco2 cells, underscoring the complex and multifaceted impact of this toxin on cellular health.

After being cultured for about 14–21 days, Caco2 cells begin to differentiate to form a monolayer with tight junctions and express a variety of enzymes and transporters, which are characteristics typical of mature enterocytes in the small intestine. Several studies have employed the use of differentiated Caco2 monolayers in studying the cytotoxic effects that exposure to palytoxins have on the absorption and transport of substances across the intestinal epithelial barrier. One such study reported that palytoxin, when applied to Caco2 monolayers that had been cultured for 21 days, led to the disruption of cell monolayers at concentrations exceeding 0.135 nM (roughly 361.2 ng/mL). Moreover, a significant decrease in cellular metabolism was observed at concentrations of 1.35 nM (approximately 3612 ng/mL) after a 24 h exposure period, as determined by the AlamarBlue assay, which measures the mitochondrial activity of viable, metabolically active cells. Interestingly, the study also reported a dose-dependent decrease in the trans-epithelial electrical resistance (TEER), a measure of ionic movement across the paracellular pathway, suggesting a disruption of the stability and integrity of the monolayer due to palytoxin exposure. However, the tight junctions of these cells appeared to remain unaffected [121]. In a separate study, researchers utilized differentiated Caco2 monolayers, 25 days post-seeding, to investigate several parameters following a 24 h exposure to either palytoxin standard or purified ovatoxin-a or -d, which were isolated from *O.* cf. *ovata*. The parameters examined included cell viability, assessed via the MTT assay; the inflammatory response, evaluated through the release of IL-8; and barrier integrity, measured by TEER. The study found that the IC_50_ value of palytoxin standard was 200 ng/mL. The release of IL-8 reached its peak at concentrations between 3.13 and 12.5 ng/mL for palytoxin standard, and was statistically significant at a concentration of 5 ng/mL for ovatoxin-a and even lower concentrations of 1.25 and 2.5 ng/mL for ovatoxin-d. In terms of barrier integrity, TEER significantly decreased with 0.25 ng/mL of palytoxin and 2.5 ng/mL of ovatoxin-a. Interestingly, ovatoxin-d did not have any observable effect on intestinal barrier integrity up to a concentration of 5 ng/mL. In general, palytoxin exhibited greater toxicity than the ovatoxins in this study, with ovatoxin-d being identified as the least toxic [61].

Upon reviewing the cytotoxic effects of palytoxins on Caco2 cells, it is evident that the reported IC_50_ values span a broad range. This variability underscores the considerable influence of both the assay methodology (such as the MTT versus CyQuant assays) and the cellular state (undifferentiated cells versus differentiated monolayers) on the observed cytotoxicity. It appears that undifferentiated Caco2 cells are more susceptible to the cytotoxic effects of palytoxins. On the other hand, differentiated Caco2 cells, which more closely resemble the intestinal epithelial barrier, generally exhibit a higher resistance to palytoxin, as indicated by their higher IC_50_ values. This highlights the critical importance of considering the cellular context when interpreting cytotoxicity data. Despite these variations, all studies consistently demonstrate the potent cytotoxic effects of palytoxin on Caco2 cells, with significant cellular changes occurring at just nanogram concentrations. Interestingly, a study that evaluated the toxicity of palytoxins across a variety of cell types found that Caco2 cells exhibited a lower sensitivity to palytoxin compared to other cell lines [117]. This finding further emphasizes the complex interplay between toxin concentration, cellular context, and observed cytotoxic effects.

While the Caco2 human intestinal cell line has provided valuable insights into the cytotoxic effects of palytoxin exposure, the toxin’s impact on other key cell types within the gastrointestinal tract has also been explored. The enteroglial cell line EGC, derived from the jejunum of an adult male rat, is a valuable tool for understanding the complex interactions between the enteric nervous system (often referred to as the “second brain”) and homeostasis of the gut. Utilizing the MTT assay following a 24 h exposure in EGC cells, one study reported an IC_50_ value of 0.39 ng/mL for palytoxin standard and comparable IC_50_ values of around 2.5 ng/mL for ovatoxins-a and -d [117].

The neurotoxic effects of palytoxins have been demonstrated across a variety of neuronal environments, encompassing both the enteric and central nervous systems. Neuro2a cells, a mouse neuroblastoma cell line widely employed as a model for studying neuronal differentiation, neurite outgrowth, and neurotoxicity, have shown extreme sensitivity to palytoxin, evidenced by the extremely low IC_50_ values and toxic effects observed in the picomolar range. Upon exposure to a sub-cytotoxic concentration of 1 pM of palytoxin standard, these cells exhibited an up-regulation of genes involved in cell survival, neuronal development, and apoptosis [122]. Three separate studies, each employing the MTT assay following 19 to 24 h exposures to palytoxin standard, have reported distinct IC_50_ values of 5 pM (0.0134 ng/mL), 43 pM (approximately 0.115 ng/mL), and 0.69 ng/mL [117,122,123]. Interestingly, these findings are in line with another study that used similar experimental conditions but with palytoxins isolated and extracted from *Trichodesmium* extracts, which contain both palytoxin and 42-hydroxy-palytoxin. This study reported a comparable IC_50_ value of 170 pM (approximately 0.456 ng/mL) [7]. Despite the extreme sensitivity to palytoxin, Neuro2a cells were found to be less sensitive to ovatoxins-a and -d, with both toxins showing similar IC_50_ values of around 2.5 ng/mL [117]. These findings not only highlight the differential cytotoxic effects of palytoxins, but also emphasize the potency of palytoxins across diverse sources, underscoring the importance of further studies to elucidate their mechanisms of action.

The cytotoxic impact of palytoxins has been demonstrated to be particularly potent on human lung cells. A549 cells, a type of human lung cell, exhibited a comparable response to both palytoxin standard and its ovatoxin-a and -d congeners, with an IC_50_ value of roughly 1.5 ng/mL after a 24 h exposure duration. When compared to a variety of other cell types, these cells were found to be second only to the Neuro2a and EGC nervous system cell lines in terms of sensitivity to palytoxin standard, and were the most sensitive cell line to ovatoxins-a and -d [117]. Interestingly, an early study that explored the effects of palytoxin isolated from *P. tuberculosa* on human bronchial epithelial cells discovered that concentrations as low as 1 pM (approximately 2.68 ng/mL) could inhibit the growth of normal human bronchial epithelial cells (NHBE), human mucoepidermoid lung carcinoma cells (HUT292), and immortalized human bronchial epithelial cells (BEAS-2B cells) [124]. Taken together, these findings underscore the remarkable sensitivity of both normal and tumorigenic human lung cells to palytoxin and its congeners, highlighting the need for further research into the mechanisms underlying this sensitivity and the potential implications for human health.

A recent study published in 2024 examined the cytotoxic effects of palytoxin and several of its ovatoxin congeners across a diverse panel of cell types. Overall, the IC_50_ values, as determined by the MTT assay, ranged from approximately 0.4 to 18 ng/mL following a 24 h exposure to palytoxin standard, ovatoxin-a, or ovatoxin-d. Among the various cell lines exposed to palytoxins, the IC_50_ values were the lowest for the nervous system cell lines Neuro2a and EGC, as well as the human lung cell line A549 [117]. Collectively, these findings reveal significant variations in cytotoxicity, with varying IC_50_ values and subsequent effects, indicating differential susceptibility of diverse cell lines to this group of marine toxins. Palytoxin and its congeners demonstrate extreme potency in certain cell types (such as neuronal and lung cells), but not others (like intestinal cells). While the differential sensitivity observed across various cell lines contributes to a comprehensive understanding of the cellular mechanisms underlying the toxicity of palytoxins, it also emphasizes the necessity for thorough in vivo studies to facilitate a transition from the microscopic view of individual cells to the macroscopic perspective of entire organisms.

## 6. Ex Vivo Studies

In 1974, initial pharmacological studies on the ex vivo toxicity of palytoxin were independently published by researchers from both the University of Oklahoma and the University of Tokyo [27,125]. While investigating the causes of ciguatera poisoning, Japanese researchers isolated *P. tuberculosa* from the digestive tract of the poisonous filefish *Alutera scripta* [27]. They were able to extract palytoxin from the isolated coral and found that at concentrations between 0.1 and 10 ng/mL, it caused a positive ionotropic effect in the atria and papillary muscle of guinea pigs, and also induced contractions in the ileum. The palytoxin also reduced muscle contractions induced by either direct or indirect stimulation in a mouse phrenic nerve hemidiaphragm assay, eventually leading to a state of contracture. The palytoxin also elicited a continuous state of contraction in frog rectus muscle, as well as depolarization in the sartorius muscle, at a concentration of 10 ng/mL [27].

The researchers at the University of Oklahoma reported comparable results after examining the toxicity of crude palytoxin isolated from *P. caribaeorum* on the smooth muscles and cardiac tissues of several animal species. Of the smooth muscles examined, exposure to 10 ng/mL of palytoxin rendered guinea pig ileum unresponsive, whereas exposure to 2 μg of palytoxin resulted in notable and irreversible vasoconstriction in isolated rat hindquarter. The coronary vasculature proved to be the most sensitive to palytoxin’s vasoconstrictive effects, however. In isolated perfused guinea pig heart, introducing 0.5 ng of palytoxin into the coronary circulation resulted in significant vasoconstriction within 5 to 10 s. Additionally, exposure to 50 ng/mL of palytoxin caused spontaneously beating rat auricles to become arrhythmic and eventually stop beating, while 100 ng/mL rendered an electrically driven guinea pig ventricular strip unresponsive to electrical stimulation [125]. In conclusion, the comprehensive studies published in 1974 revealed the profound physiological impacts of palytoxin exposure, underscoring its potential as a significant health risk and emphasizing the importance of continued research into its mechanisms of action and potential countermeasures.

## 7. In Vivo Studies

### 7.1. Acute Toxicity of Palytoxin

Some of the earliest pharmacological investigations into the toxicity of palytoxin were serendipitously shifted in focus from the quest to understand ciguatera food poisoning. While distinct from both maitotoxin and ciguatoxin, palytoxin can elicit similar symptoms due to its action on ion pump systems and consequent impairment of nerve conduction [126,127]. One of the first studies assessing the in vivo toxicity of palytoxin was published in 1969. Japanese researchers, while investigating the causes of ciguatera, discovered the presence of *P. tuberculosa* in the viscera and gut contents of a toxic filefish, marking the first proposition of a food chain hypothesis for the bioaccumulation of palytoxin [26]. Ethanolic extractions were purified from both the ingested materials found in the gut of the filefish specimens and from procured samples of *P. tuberculosa*. These purified extracts were then administered to mice via intraperitoneal (IP) injections. The preparation was found to be toxic at a concentration of 0.06 μg/g (approximately equivalent to 60 μg/kg), with the lethal dilution, which resulted in the death of three mice within a period of 30 min to 2 h, corresponding to mouse units of 4.5 and 2.5, respectively. The clinical signs induced in the mice included a decrease in activity, hypersalivation, hind limb paralysis, and dyspnea culminating in death [26]. A few years later, in 1971, Moore and Scheuer from the University of Hawaii observed similar clinical signs in mice following administration of a crude sample of palytoxin isolated from the zoanthids of the legendary Hawaiian tidepool (later determined to be *P. toxica*). They reported a much lower LD_50_ of 400 ng/kg via IP injection and 150 ng/kg through intravenous (IV) injection [21]. Shortly thereafter, in 1973, researchers at the University of Tokyo isolated and purified palytoxin from *P. tuberculosa* and reported that the preparation was lethal to mice at a minimum dose of 600 ng/kg via IP injection [47].

Several preliminary toxicological studies on palytoxin toxicity were published in 1974, offering further perspectives in a variety of animal species [27,125,128]. Palytoxin isolated from *P. mammillosa* was administered via IP injection by the research group at the University of Oklahoma, who reported an LD_50_ between 50 and 100 ng/kg in male Swiss-Webster mice [125]. Researchers at the University of Tokyo used palytoxin isolated from *P. tuberculosa* and reported an IV LD_50_ of 530 ng/kg in mice and 200 ng/kg in cats [27]. Remaining one of the most comprehensive toxicological evaluations of palytoxin to date, researchers at the U.S. Army Toxicology Division also published their findings in 1974 on the acute toxicity of palytoxin in several animal species by various routes of administration [128]. They used crude toxin extracted from samples of *P. vestitus* obtained from the University of Hawaii (suspected to be *P. toxica*). The acute toxicity of palytoxin was examined in several species: rhesus monkey (*Macaca mulatta*), dog (*Canis familiaris*), guinea pig (*Cavia porcellus*), New Zealand white rabbit (*Oryctolagus cuniculus*), Wistar–Sprague Valley rat (*Rattus norvegicus*), and ICR white mouse (*Mus musculus*). These species were administered the toxin via various routes: intravenous (IV), intramuscular (IM), subcutaneous (SC), intraperitoneal (IP), intratracheal (IT), intragastric (IG), intrarectal, intradermal, percutaneous, and ocular [49,82,128]. In this study, palytoxin appeared to be most toxic when administered intravenously. The rabbit, dog, and monkey were the most susceptible to its toxic effects, with reported 24 h LD_50_ values of 25, 33, and 78 ng/kg, respectively [128]. The IV LD_50_ values of rats, guinea pigs, and mice were slightly higher, with values of 89, 110, and 450 ng/kg, respectively [128]. Palytoxin was also shown to be lethal by IM and SC injections, as well as IT instillation, with these routes of administration showing similar LD_50_ values in the rat (240, 400, and 360 μg/kg, respectively) [128]. Dermal and ocular exposures to palytoxin were not lethal but did produce toxic irritant effects [128]. When the outcome was lethal, the causes included renal failure accompanied by shock and uremia, widespread hemorrhagic diathesis with necrotizing vasculitis and ischemia, congestive heart failure, hemorrhagic pneumonitis, and simultaneous or secondary infections [128].

Replicating the comprehensive toxicological research detailed in this original study has been an ongoing process. Subsequent studies have further expanded our understanding by providing reports on the median lethal doses and clinical observations following the administration of purified palytoxin and related compounds across a range of animal models. It is noteworthy that the reported LD_50_ estimates for the intravenous toxicity of palytoxin exhibit considerable variation, likely attributable to differences in palytoxin preparations and observation times, yet among the tested species, mice appear to be the least susceptible to this route of administration [21,27,75,128]. For example, in a 1992 study, putative palytoxin isolated from the sea anemone *Radianthus macrodactylus* was reported to have a 24 h IV LD_50_ value of 740 ng/kg in male white mice (strain not specified). The identity of the extracted palytoxin was confirmed through spectroscopy by comparing the obtained data with previously published articles as opposed to the use of a palytoxin standard [75]. Additional studies using multiple exposure modalities have provided a more comprehensive picture of the toxicological profile of palytoxin, its analogues, and congeners.

An early study reported a 24 h LD_50_ of 500 ng/kg for palytoxin isolated from *P. tuberculosa* following IP injection into female Swiss CD-1 albino mice [129]. Similarly, while investigating a fatal case of clupeotoxism, a type of human intoxication from the consumption of clupeoid fish, researchers extracted palytoxin from frozen samples of the sardine species *Herklotsichthys quadrimaculatus* implicated in the poisoning. When injected intraperitoneally into male ddY mice, an LD_50_ of 450 ng/kg was reported [60]. However, a later study reported a slightly higher LD_50_ of 720 ng/kg for palytoxin administered intraperitoneally in Swiss albino mice (sex not specified) [130]. In 2008, researchers aimed to standardize the mouse bioassay protocol for palytoxin. They investigated and documented a set of distinctive initial clinical signs following IP injection of the commercially available palytoxin standard in male Swiss mice. The 24 h LD_50_ was reported as 295 ng/kg, with mice showing characteristic clinical signs that included extension of the hind limbs, concave curvature of the spine, bristling fur, potential loss of sight, seizures, and labored breathing within 15 min of injection [131]. Corroborating the slightly higher LD_50_ values observed in earlier research, a study published in 2020 reported an LD_50_ value of 680 ng/kg for IP administration of palytoxin standard in female Swiss mice [64]. In this study, blood biochemistry analysis showed significant elevations in creatine kinase beginning at doses of 220 ng/kg, indicating the presence of severe skeletal muscle damage. There were also significant reductions in blood sodium and elevations in blood potassium at palytoxin doses of 680 ng/kg and higher [64].

Although the reported clinical observations are similar, analogues and congeners of palytoxin have been shown to exhibit reduced potency, as evidenced by higher LD_50_ values, indicating a lower toxicity compared to the parent compound. Ostreocin-d, identified as the major palytoxin congener in *O. siamensis*, has a reported LD_50_ of 750 ng/kg when administered to mice (strain and sex not specified) via IP injections [2]. Similarly, another study reported an LD_50_ of 720 ng/kg resulting from IP injections of a “palytoxin-like material” extracted from cultured *O. siamensis* cells in female Swiss mice. These mice initially exhibited a swaying gait, followed by extension of the hind limbs and cessation of spontaneous movement within 20–30 min. Respiration rates subsequently decreased, punctuated by intermittent gasps, which culminated in death after approximately 40 min [132]. An even higher dose of 5 μg/kg of ostreocin-d, isolated and purified from *O. siamensis*, proved lethal to male ICR mice. This led to clinical signs such as limb paralysis, systemic congestion, and stomach and intestinal erosion [133]. Similarly, male ICR mice given an IP dose of 1.5 μg/kg of palytoxin extracted from the crab *Demania alcalai* resulted in fatality. This was accompanied by erosion of the stomach and small intestine [127].

Building upon studies on the IP toxicity of palytoxins, in 2018, the U.S. Army Medical Research Institute of Infectious Diseases published a study that examined the toxicity and pathophysiology of palytoxin and several of its congeners following IP administration in female Fischer F344 rats. The four studied preparations of palytoxin included the commercially available palytoxin standard isolated from Japanese *P. tuberculosa*, 42-OH-palytoxin isolated from Hawaiian *P. toxica*, a 50:50 mix of palytoxin and 42-OH-palytoxin obtained from Hawaiian *P. tuberculosa*, and ovatoxin-a isolated from cultures of *O. ovata*. The reported LD_50_’s for IP injections ranged approximately from 1 to 3 μg/kg: 1.81 μg/kg for palytoxin standard, 1.93 μg/kg for 42-OH-palytoxin, 0.92 μg/kg for the 50:50 mix, and 3.26 μg/kg for ovatoxin-a. In the rats that survived for at least 20 h post-injection, the intraperitoneally administered palytoxins induced histologic lesions in the lungs, liver, kidneys, adrenal glands, and salivary glands. The authors also reported observations of acute cardiomyocyte degeneration due to hypercontraction and myocardial inflammation [49].

Intratracheal (IT) administration serves as a practical alternative to aerosolization, enabling researchers to deliver substances directly into the lungs of experimental animals with precision and control. This method facilitates the examination of respiratory effects while avoiding the complexities and expenses of inhalation exposure systems. A study compared the toxicological effects of palytoxin standard and ostreocin-d, isolated from *O. siamensis*, via IT administration in male ICR mice and male Wistar rats. A dose of 2 μg/kg of palytoxin caused hemorrhaging, paralysis, and death within 2 h in the mice. Autopsy revealed extensive alveolar bleeding and destruction, edema around lung vessels, gastrointestinal erosion, and glomerular atrophy in the kidneys. Ostreocin-d proved lethal within 1 h at doses of 13 μg/kg and within 6 h at doses of 11 μg/kg in these mice. Bleeding was observed in the lungs and stomach, but no histological changes were noted in the kidneys. In the rats, the lethal effect of palytoxin was observed at doses ranging from 5 to 7.5 μg/kg, while for ostreocin-d it was observed at 11 μg/kg [133]. Ostreocin-d demonstrated lower potency compared to palytoxin, even when administered intratracheally.

Compared to IT administration, the administration of palytoxins by aerosol appears to be significantly more potent, with LD_50_ values in the nanogram range. Only one study has reported on the acute toxicity of palytoxins resulting from inhalational exposures. Various palytoxin preparations administered to female Fischer F334 rats via nose-only aerosol exposure resulted in LD_50_ values ranging from 31 to 63 ng/kg (41 ng/kg for palytoxin standard, 45 ng/kg for 42-OH-palytoxin, 63 ng/kg for the 50:50 mix, and 31 ng/kg for ovatoxin-a). Upon necropsy, the animals exhibited gross lesions primarily in the lungs and liver. Additional findings included pulmonary and hepatic congestion, consolidation of pulmonary lobes with hemorrhagic areas, sporadic epistaxis (nosebleeds), and red crusting around the eyelids and nares [49].

The oral administration of palytoxins has been reported to be far less toxic than other routes of administration. In the preliminary toxicity study, intragastric administration of up to 10 μg/kg of palytoxin in male Wistar–Sprague Valley rats resulted in minimal to no toxicity. Furthermore, doses as high as 40 μg/kg resulted in the death of only one out of four rats, leading researchers to conclude that the LD_50_ of this route of administration must be greater than 40 μg/kg [128]. A subsequent study reported an LD_50_ of 767 μg/kg for palytoxin isolated from *P. tuberculosa* and administered via gavage to female Swiss CD-1 mice. Histological analyses showed acute inflammation of the forestomach and changes in the liver and pancreas in these animals. Additionally, transmission electron microscopy revealed ultrastructural alterations in the cardiac and skeletal muscle cells [134]. Another study administered palytoxin standard to female Swiss mice via oral gavage and reported an LD_50_ of 599 μg/kg [64]. Similarly, in Swiss albino mice (sex not specified), the LD_50_ for gavage administration of palytoxin was 510 μg/kg and exceeded 2500 μg/kg when administered by voluntary food intake [130]. The relatively low oral toxicity of palytoxin is likely due to its limited absorption in the gastrointestinal tract. However, it is important to note, as highlighted by another researcher, that the distinctive structure of the rodent stomach must be considered when assessing acute oral toxicity via gavage administration [135].

In vivo studies have helped to highlight the significant toxicological threat posed by palytoxins, with LD_50_ values often measured in the nanogram range (Appendix A). Despite variations in these reported values, which can be attributed to different routes of administration, the animal model used, and the methods of palytoxin isolation and preparation, the extreme potency of palytoxin, as well as its analogues and congeners, is consistently evident. The effects of exposure to sublethal concentrations of palytoxins in in vivo models have been investigated. This research provides a pivotal window into the nuanced interactions of this potent toxin with various organ systems, offering valuable insights into its systemic impact and the underlying mechanisms of action.

### 7.2. Sublethal Effects of Palytoxin

One of the earliest investigations into the effects of sublethal doses of palytoxin was conducted by researchers at the University of Oklahoma in 1974. Their study focused on the significant cardiovascular impact resulting from exposure to *P. caribaeorum* in anesthetized dogs. Within 30 s of administering a 100 ng/kg IV dose of palytoxin, the dogs exhibited tachycardia, altered pulsatile blood pressure, and a hypotensive response, with arrhythmia following within 3 min. When a higher IV dose of 500 ng/kg was administered, the animals displayed erratic and atypical blood pressure patterns. This led to cardiac arrest within 4 to 6 min and subsequent respiratory failure within 10 min [125]. The results of this study led the researchers to propose that the primary cause of lethality from palytoxin is likely profound coronary vasoconstriction. Supporting this notion, a study published the following year in 1975 reported that the vasodilators papaverine or isosorbide dinitrate effectively reversed the cardiotoxic effects of the IV administration of 10 ng/kg of palytoxin extracted from Hawaiian *P. vestitus* in dogs. To achieve this, the therapeutic injections were directly delivered into the left ventricle of the heart at the onset of cardiac clinical signs, as venous blood flow rapidly stagnated due to the toxin [136]. A few years later, another study further investigated the cardiovascular toxicity of palytoxin in canines. Administering a sublethal intravenous dose of 40 ng/kg of palytoxin isolated from *P. tuberculosa* resulted in the constriction of coronary, femoral, and renal arteries, leading to an increase in total peripheral resistance and an abrupt decrease in cardiac output [137]. Researchers propose that the toxicity of palytoxin arises from severe vasoconstriction throughout the body, particularly in the coronary and renal vascular beds, coupled with a reduction in cardiac function.

The IP administration of sublethal doses of palytoxins in in vivo animal models has broadened our knowledge of the toxin’s impacts. One study reported a No-Observed-Adverse-Effect-Level (NOAEL) of 6.8 ng/kg for palytoxin standard following IP administration in female Swiss mice. When the dosage was increased to 22 ng/kg or more, the mice began to exhibit clinical signs of distress such as abdominal pain, hind limb stretching, ataxia, and lethargy. At sublethal doses of 220 ng/kg and above, there was a significant increase in blood creatine kinase levels. Animals at this dose exhibited lethargy, abdominal pain, and hind limb paralysis. The IP LD_50_ was reported to be 680 ng/kg in this study [64]. Another study examined intestinal injuries resulting from experimental sublethal palytoxin poisoning. Male ICR mice were given an IP dose of 1 μg/kg of palytoxin extracted from the crab *Demania alcalai* (a dose of 1.5 μg/kg was lethal in this study). Within 16 h, the mice exhibited edema, necrosis of the lamina propria within the villus of the small intestine, damage to lymphoid tissues, severe peritonitis, and diarrhea [127]. In a subsequent study, it was found that an IP injection of a sublethal dose of 4 μg/kg of ostreocin-d, isolated and purified from *O. siamensis*, led to mild diarrhea in male ICR mice approximately 10 h post-administration. Recovery and an increase in body weight were observed within 24 h [133]. Despite causing similar clinical signs, ostreocin-d was shown to be far less potent than palytoxin.

It is known that consuming seafood contaminated with palytoxins, such as reef fish and crabs, can lead to severe gastrointestinal symptoms, including abdominal cramping and profuse watery diarrhea. Building on this understanding, research studies have focused on the acute and chronic effects of sublethal administration of palytoxins via gavage, aiming to elucidate the specific physiological responses and health risks associated with oral exposure. Acute studies typically involve a one-time exposure to the toxin, allowing for the observation of immediate toxic effects. In contrast, chronic studies involve repeated exposures over an extended period, enabling the observation of cumulative effects and potential long-term damage. In an acute oral toxicity study where palytoxin isolated from *P. tuberculosa* was administered via gavage to female CD-1 mice, the NOAEL was estimated at 300 μg/kg following a 24 h observation period, with an LD_50_ of 767 μg/kg [134]. However, a more recent study reported a NOAEL that was 10 times lower in female Swiss mice receiving a single oral dose of palytoxin standard by gavage. In this study, clinical signs of pain were exhibited at a dose of 36 μg/kg, while no observable adverse effects were noted at doses of 15 μg/kg. Oral administration of the toxin at doses of 36 μg/kg and above resulted in significant increases in blood levels of alanine transaminase, aspartate transaminase, lactate dehydrogenase, and creatine kinase. At a dose of 300 μg/kg, significant increases in blood sodium and chloride levels were observed. The LD_50_ for orally administered palytoxin was 599 μg/kg in this study [64].

Similarly, another study identified the Lowest-Observed-Adverse-Effect-Level (LOAEL) as a dose of 30 μg/kg/day in female CD-1 mice administered palytoxin standard via gavage over a treatment period of 7 days [63]. The primary organs affected included the lungs, heart, and gastrointestinal tract. Specifically, the lungs exhibited severe inflammation; the heart showed signs of eosinophilia and separation of fibers within the myocardium; and the gastrointestinal tract presented gastric ulcers and an accumulation of fluid in the intestines. The NOAEL was estimated to be 3 μg/kg/day in this study, which is about 100 times lower than the estimate from a previous study [63,134]. Another study reported on the chronic oral toxicity of palytoxin standard after gavage administration in female Swiss mice, but over a longer 28-day period. The LD_50_ was 440 ng/kg, and the NOAEL was found to be 30 ng/kg for repeated daily oral administration [114]. Chronic daily exposure to the toxin had profound effects on the blood biochemistry of the mice. For instance, blood levels of alanine transaminase significantly increased in animals dosed with 100 ng/kg of palytoxin. Statistically significant differences were also observed in blood sodium, potassium, and chloride levels. Sodium levels decreased at a dose of 300 ng/kg of palytoxin, and potassium levels were elevated in the animals fed daily between 30 and 300 ng/kg of palytoxin. Blood chloride levels were elevated with palytoxin doses between 30 and 1000 ng/kg. This dysregulation of electrolyte levels aligns with alterations previously reported from a single oral dose of palytoxin [64]. The toxin also caused macroscopic alterations in the digestive tract as well as ultrastructural alterations in the stomach. Upon necropsy, animals chronically exposed to the toxin exhibited abdominal swelling, an absence of solid content, and the presence of gas and mucus in the stomach and intestines [114].

As evidenced in these studies, there are significant variations in the NOAEL estimates following the oral administration of palytoxin. These discrepancies are likely due to the different methodologies employed in the studies, specifically the contrast between acute, single-dose administration and chronic, repeated administration over several days, as well as the different animal strains used and the varying degrees of purity of the toxin preparations administered. As a result, the NOAEL estimates from these two types of studies can vary significantly, reflecting the diverse range of effects that palytoxins can exert under different exposure scenarios.

Both ostreocin-d and palytoxin standard have been shown to produce comparable toxicological effects after gavage administration of sublethal doses in male ICR mice. In mice given 200 μg/kg of palytoxin or 500 μg/kg of ostreocin-d, mild erosion, slight accumulation of fluid in the stomach, and minor injuries to the small intestines, lungs, and kidneys were observed at 2 h post-administration. However, these clinical signs were no longer present after 24 h, suggesting a temporary effect of these substances on the mice [133]. Similarly, sublingual administration of either palytoxin standard or ostreocin-d, in doses ranging from 172 to 235 μg/kg, led to comparable organ alterations as well as similar timelines for these changes. In the hours following administration, the mice exhibited reduced activity and an increased respiration rate. Microscopic examination at autopsy revealed minor bleeding in the lungs, observed at 30 min, followed by inflammation of the interstitial tissue between 1.5 and 8 h, and alveolar destruction at 8 to 24 h. Additionally, there was edema in the tissues of the gastrointestinal tract along with exposure of the gastric glands and erosion of the small intestinal villi after 30 min. Within 8 to 24 h, the kidneys exhibited signs of glomerular atrophy. Nonetheless, the alterations induced by ostreocin-d were less severe compared to those caused by palytoxin, suggesting a difference in the toxicity of these two substances. Taken together, these results indicate that palytoxin and ostreocin-d penetrate the bloodstream via the oral mucosa, subsequently reaching various organs to induce organ-specific pathological changes [133].

The effects of sublethal doses of either palytoxin standard or ostreocin-d, administered intratracheally to male ICR mice, were also examined. Intratracheal delivery of a sublethal dose of 1 μg/kg palytoxin standard led to a temporary paralysis that lasted between 1 and 2 h, followed by a recovery phase. Despite appearing normal after 24 h, these mice exhibited injuries to multiple organs, including pulmonary emphysema, edema in the gastrointestinal tract, and glomerular atrophy. Mice that received an IT delivery of sublethal doses of ostreocin-d (ranging from 4.5 to 9 μg/kg) exhibited a squirming gait and slightly paler eye color within 2 h. After 24 h, their body weight had decreased, there was evidence of bleeding in the lungs, and the proximal section of the small intestine appeared reddish and semi-transparent [133].

In conclusion, the studies discussed in this review highlight the diverse range of toxicological effects that can result from exposure to palytoxins. These findings underscore the importance of ongoing research into the mechanisms of action of these potent toxins, as well as the development of effective strategies for mitigating their impact on human and animal health. The variations in observed effects and toxicity levels across different routes of administration and exposure scenarios further emphasize the complexity of palytoxin toxicity and underscore the need for comprehensive, multifaceted approaches to studying these compounds. The exploration of the toxicity of palytoxins in living organisms has provided a clear indication of the potency of this highly toxic substance, with LD_50_ values frequently cited in the nanogram range across multiple routes of administration. However, the existing literature on bioassays for palytoxins lacks clarity in various crucial aspects, as there are notable discrepancies among the reported median lethal doses and NOAELs for various species, routes of administration, and durations of observation in the animal models used. Nonetheless, while there have been significant strides in the detection and assessment of palytoxins, the literature reflects a need for greater clarity and standardization in these key areas.

## 8. Future Research

It is clear that palytoxins are toxicologically significant due to their potent bioactive properties. Their toxicity has been well documented through various in vitro and in vivo investigations, with reported LD_50_s in the nanogram and microgram range underscoring their potency. Palytoxins pose a risk to humans via multiple exposure pathways, including ingestion, inhalation, and dermal contact. Moreover, these toxins can be encountered through a variety of vectors, such as fish, shellfish, zoanthids, and even marine waters during *Ostreopsis* blooms. Palytoxins have been implicated in various forms of seafood poisoning, notably clupeotoxism, Haff disease, and ciguatera poisoning [60,76,84,85,116]. Their ability to transfer and bioaccumulate in marine food webs not only heightens the potential for human exposure but also highlights the need for comprehensive monitoring and management strategies in marine ecosystems. Certain zoanthids, particularly the highly toxic variety of *Palythoa* spp., are readily available in retail commerce and found in home aquariums. However, owners are often unaware of the risk of exposure to these deadly toxins [3,53,81]. These associations underscore the broad impact of palytoxin on human health and emphasize the importance of ongoing research into its mechanisms of action, detection methods, and potential therapeutic strategies. There is also a need for public education and regulation to prevent and manage the health risks associated with accidental palytoxin exposure.

It has been unequivocally demonstrated that palytoxin and its various analogues possess a strong potential for toxicity in humans and animals, making them a major concern for public health. However, the full toxicological potential of palytoxin compounds remains to be fully evaluated. In fact, the confirmation and quantification of palytoxins in case reports are often lacking, potentially due to limitations of biochemical assays used for palytoxin detection [20,81,138,139].

To address gaps in our understanding of palytoxin toxicity and to develop effective strategies for risk assessment and management, future research efforts on palytoxins should focus on several key areas. These include elucidating the molecular mechanisms of palytoxin-induced cytotoxicity; identifying specific molecular targets involved in their toxic effects; developing certified reference materials and sensitive, specific, and reproducible methods for the accurate detection and quantification of palytoxins in environmental samples, seafood products, and biological matrices; assessing the long-term health effects of palytoxin exposure including the evaluation of the potential carcinogenicity and genotoxicity; exploring novel approaches for the prevention and treatment of palytoxin poisoning, including the development of antivenins and supportive therapies; and monitoring and predicting the occurrence of palytoxin-producing algal blooms in marine environments and implementing effective management strategies to mitigate their impact on human health and marine ecosystems. By addressing these areas with thorough and rigorous scientific research, our understanding of palytoxin toxicity can be enhanced, which will ultimately accelerate strategies to protect public health and the environment from the hazards associated with these potent marine toxins.

## Data Availability

No new data were created or analyzed in this study. Data sharing is not applicable to this article.

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
