# Peer review of "History and Toxinology of Palytoxins"

_toxins, 2024, doi:10.3390/toxins16100417_

Round 1

Reviewer 1 Report

Comments and Suggestions for Authors

The manuscript reviews the knowledges on palytoxins and some of the ovatoxins available so far.  

The cited papers range from very old (1970-80) to very recent (2020), sometimes missing some papers published in between: in particular, the Authors often emphasize the results of some very old papers, which, due to the analytical methods of the time, could have been obtained by working with palytoxin of scarce purity.

In the manuscript, the in vitro results are reported and highlighted before the in vivo results: it is known that the in vitro results are important to explain the observed in vivo toxic effects. The Authors speculate about differences in the toxic effects of these compounds based on their in vitro effects on different cell lines (e.g. skin-derived cell lines, respiratory cell lines, etc.). I suggest modifying the order of the paragraphs, discussing the data obtained from testing in vivo before those obtained by testing in vitro

In general, data obtained from in vitro studies are often discussed together with data obtained from ex vivo studies, as if the two methods were one and the same.

The Authors often reported the in vitro data as if they were central to comparing and understanding differences in the toxicokinetics and in the risk assessment of palytoxins («In vitro studies are… an invaluable asset to our understanding of the respective health implications resulting from various routes of exposure to palytoxins» lines:  242-245), when in fact in vitro data explain the mechanism of action of the toxins, and thus contribute to describing the corresponding toxicokinetics and risk assessment; in vitro data cannot, however, replace in vivo data, without which there can be no proper insight into the risk assessment and toxicokinetics of any compound, because in vitro data obtained using cell lines represent a very simplified system in which toxic compounds have to cross only a few cell membranes, and as such are not representative of the problems that would arise in a complex system such as the human body.

Minor Modifications:

Keywords: please change ovatoxin with ovatoxins, please eliminate zoantides

and add oral exposure.

Lines 344-349 and 486-487: More research is needed to fully understand the role of palytoxins in Ciguatera. Please better emphasize  the “non-involvement” of palytoxin in Ciguatera poisoning. 

Line 513: pharmacological studies, please change in toxicological studies

Line 529:   please change In this study  with In these studies

Line 732-738: The authors have to mention  the variability of the results can be due, not only to the different animal species and strains used, but also to the different purity of the toxins used,  despite of their commercial origin.

Line 789:  Extreme potency: please delete Extreme leaving only potency.

Author Response

September 4, 2024

EDITOR

Toxins Journal

Manuscript ID: toxins-3163534

History and Toxinology of Palytoxins

RE:  Response of Authors to reviewers comments

Dear Editor:

The following provides specific responses to each reviewer comment provided, with the original verbatim comment italicized and the response of the authors addressing the specific comment is intended and follows each comment. 

Author Response to Reviewer 1:

I suggest modifying the order of the paragraphs, discussing the data obtained from testing in vivo before those obtained by testing in vitro. 

Authors justified moving review subject matter from in vitro to ex vivo to in vivo studies as logical sequence for creating a logical flow of information, allowing presentation of the literature from the most controlled and simplified systems to the most complex and realistic ones.

In general, data obtained from in vitro studies are often discussed together with data obtained from ex vivo studies, as if the two methods were one and the same.

The portion describing the ex vivo studies were removed from the In vitro Studies section and placed on its own subsection. 

Response, pg. 2

Keywords: please change ovatoxin with ovatoxins, please eliminate zoanthids and add oral exposure.

Authors replaced ovatoxin with ovatoxins; added oral exposure; the term ‘zoanthids’ was not eliminated as these species are key to this study and would potentially weaken searches for reviews on toxins associated with this species.

Lines 344-349 and 486-487: More research is needed to fully understand the role of palytoxins in Ciguatera. Please better emphasize the “non-involvement” of palytoxin in Ciguatera poisoning.

Authors thank reviewer for identifying this detail.  Authors added details about ciguatera poisoning which is provided in the new ‘Bioaccumulation’ section.

Line 513: pharmacological studies, please change in toxicological studies

Terms changed; currently found on Line 647 of the revised manuscript.

Line 529:   please change In this study with In these studies

Currently Line 663; rather than changing verbiage, authors choose to remove extra reference so only a singular reference is identified, thus “In this study…”

Line 732-738: The authors have to mention the variability of the results can be due, not only to the different animal species and strains used, but also to the different purity of the toxins used, despite of their commercial origin.

Authors thank reviewer for identifying this detail.  Added. Lines 875-876.

Line 789:  Extreme potency: please delete Extreme leaving only potency.

Authors thank reviewer for identifying this detail.  Changed; Line 929.

Reviewer 2 Report

Comments and Suggestions for Authors

Please see attachment,

Author Response

September 4, 2024

EDITOR

Toxins Journal

Manuscript ID: toxins-3163534

History and Toxinology of Palytoxins

RE:  Response of Authors to reviewers comments

Dear Editor:

The following provides specific responses to each reviewer comment provided, with the original verbatim comment italicized and the response of the authors addressing the specific comment is intended and follows each comment. 

Author Response to Reviewer 2:

The authors are suggested to eliminate the term “marine biotoxins” throughout the text and replace it with “marine toxins”, since biotoxins is redundant.

Authors thank reviewer for identifying this oversight.  Changed throughout the text.

Please replace the term “symptoms” used in animal physiopathology for the effect of PLTXs with “clinical signs” throughout the text.

Authors thank reviewer for identifying this oversight.  Changed throughout the text.

Develop a timeline of the main research and scientific advances on the study of PLTX and analogues.

This is an excellent suggestion, however the authors felt that the systematic way that this review is presented allows the reader to follow the progress of research discovery and scientific advance through summarized presentation of data. In addition, the authors believe that an illustrative timeline would be redundant with the writing and style of the review and detract from concise presentation style that we have written this manuscript. 

Include a figure of the molecular structures of palytoxin, ostreocins, ovatoxins and mascarenotoxins, indicating the differences. Briefly add information on some of their physicochemical properties. (In the text of this section, add a short paragraph on the structure-activity relationship.)

This is an excellent suggestion, however the focus of this review as pursued by the authors is intended to detail the biological and physiological effects of palytoxins on living systems (in vitro and in vivo) and not necessarily highlight the structural characteristics of the toxins identified in this review.  After careful consideration, the authors felt that a concise description of the important hallmarks of palytoxin and homologues as written were more fitting in a review whose focus is biological endpoints associated with living systems exposure.  Therefore, we opted for description rather than illustrative comparative structural analysis. 

Include in the section on the “mechanism of action” a figure that includes a diagram or model of the main mechanisms of action by the different routes of exposure in human and animal health (e.g. oral, inhalation, topical, ophthalmic, etc.)

This is an excellent suggestion, however the focus of this review as pursued by the authors is on the biological and physiological effects of palytoxins on living systems (in vitro and in vivo) and not the molecular aspects of the MoA by exposure route.  Addition of the suggested illustrations would lengthen an already very lengthy manuscript, and although very important (as described) in the review, is peripheral to the main thrust of the writing.

Other important aspects in toxinology are the description of its etiology, its potential ecological role for the same producing species, biosynthesis and biotransformation in food chains.

Authors thank reviewer for identifying the addition of relevant sections to the review. Accordingly, the authors have added sections on Biosynthesis and Bioaccumulation, placed after Structural Analysis.

The proliferation of Ostreopsis species in aquariums and ponds has also been described, with negative effects on organisms confined in these systems, which represents a potential risk to animal and human health and to operating personnel.  What methods are suggested as the most useful for its detection and quantification?  Do any of the palytoxin analogues have any therapeutic potential? -If possible, include a brief paragraph.

Authors thank the reviewer for this important aspect of the review.  However, providing methodology details on detection and quantitation, and also review of the therapeutics would detract from the intended focus of this review, which is the biological and physiological effects of Palytoxin exposure on living systems. 

Reviewer 3 Report

Comments and Suggestions for Authors

This paper represents a welcome contribution to the knowledge of palytoxins regarding the structure and in vitro and in vivo effects on various experimental animals by different exposure routes. However in my opinion the paper would be significantly improved  if the following is taking into the consideration:

-little is said about the producers, vectors and accumulation of palytoxins and similar compounds in food webs. It would be nice to add this type of information

- at least some contents of the manuscript should be presented in the form of table and pictures. True, one table is mentioned in the text, but at least in my review documentation was not included in the review. It would be nice to present the binding of the palytoxin to Na+/k+ ATPase graphically since such illustrations are very rare in the existing literature.

- the link between clupeoism and palytoxin should be explained in some more details. Namely this phenomena could be deadly for humans, therefore some dose response relations should be mentioned, the potential producers and other vectors that cause accumulation of palytoxin or palytoxin related compounds in Clupeoformes fish and mackarels.

- the authors should explain how palytoxins might be involved in ciguatera poisoning. They should explain this. Namely, by my best knowledge neither maitotoxin nor palytoxins are directly involved in ciguatera poisoning (if at all). If evidence exists to prove their involvement this should be explained in the text. There are reported cases of human palytoxin poisoning after ingesting of seafood (mainly crabs). This should be mentioned. In this regard at least some discussion would be welcome about the possible mechanism of resistance of vector organisms to palytoxin. 

-what are ecological conditions that triggers the production of palytoxins and similar compounds in Palythoa species, cyanobacteria or dinoflagellates and what could be the physiological role of these compounds.

Specific: abbreviations like IV, SC, IM etc. should be written i.v., s.c., i.m. etc. except if this is the required journal format.

Author Response

September 4, 2024

EDITOR

Toxins Journal

Manuscript ID: toxins-3163534

History and Toxinology of Palytoxins

RE:  Response of Authors to reviewers comments

Dear Editor:

The following provides specific responses to each reviewer comment provided, with the original verbatim comment italicized and the response of the authors addressing the specific comment is intended and follows each comment. 

Author Response to Reviewer 3:

Response, pg. 4

little is said about the producers, vectors and accumulation of palytoxins and similar compounds in food webs. It would be nice to add this type of information

Added sections on Biosynthesis and Bioaccumulation, placed after Structural Analysis.

at least some contents of the manuscript should be presented in the form of table and pictures. True, one table is mentioned in the text, but at least in my review documentation was not included in the review. It would be nice to present the binding of the palytoxin to Na+/k+ ATPase graphically since such illustrations are very rare in the existing literature.

Authors thank the reviewer for this important aspect of the review. Authors agree with reviewer sentiment and have prepared an extensive table for this purpose, which is found in the supplemental data. A graphical illustration of the binding of palytoxin to the Na+/K+ ATPase is an excellent suggestion; however, the focus of this review, as pursued by the authors, is on the biological and physiological effects of palytoxins on living systems (in vitro and in vivo) and not the structural characteristics and MoA of the toxins identified in this review.  After careful consideration, the authors felt that a concise description were more fitting in a review whose focus is biological endpoints associated with living systems exposure.  Therefore, we opted for description rather than illustrative comparative structures. 

the link between clupeoism and palytoxin should be explained in some more details. Namely this phenomena could be deadly for humans, therefore some dose response relations should be mentioned, the potential producers and other vectors that cause accumulation of palytoxin or palytoxin related compounds in Clupeoformes fish and mackarels.

Authors thank reviewer for identifying this detail.  The authors agree with the reviewer, and this is described in added section entitled Bioaccumulation.

the authors should explain how palytoxins might be involved in ciguatera poisoning. They should explain this. Namely, by my best knowledge neither maitotoxin nor palytoxins are directly involved in ciguatera poisoning (if at all). If evidence exists to prove their involvement this should be explained in the text. There are reported cases of human palytoxin poisoning after ingesting of seafood (mainly crabs). This should be mentioned. In this regard at least some discussion would be welcome about the possible mechanism of resistance of vector organisms to palytoxin. 

Authors thank reviewer for identifying this detail.  The authors have described in added section entitled ‘Bioaccumulation’ in the revision.

what are ecological conditions that triggers the production of palytoxins and similar compounds in Palythoa species, cyanobacteria or dinoflagellates and what could be the physiological role of these compounds.

 Authors thank reviewer for identifying this detail.  Described in added section entitled Biosynthesis.

Specific: abbreviations like IV, SC, IM etc. should be written i.v., s.c., i.m. etc. except if this is the required journal format.

Authors thank review for identifying this detail.  Grammar has been corrected.